# Norwegian Consumers’ Skepticism towards Smoke-Flavoring of Salmon—Is It for Real?

**DOI:** 10.3390/foods11142170

**Published:** 2022-07-21

**Authors:** Lene Waldenstrøm, Marte Berg Wahlgren, Åse Strand, Jørgen Lerfall, Mari Øvrum Gaarder

**Affiliations:** 1Department of Biotechnology and Food Science, NTNU-Norwegian University of Science and Technology, NO-7491 Trondheim, Norway; marte.b.wahlgren@ntnu.no (M.B.W.); ase.strand@ntnu.no (Å.S.); jorgen.lerfall@ntnu.no (J.L.); 2Department of Sensory, Consumer and Innovation, Nofima AS, NO-1430 Ås, Norway; mari.gaarder@nofima.no

**Keywords:** Atlantic salmon, smoke-flavoring, consumer, attitude, quantitative, qualitative, survey, focus groups, curiosity, established beliefs

## Abstract

The aim of the present study was to explore Norwegian consumers’ attitudes toward smoke-flavoring of cold smoked salmon (CSS), by conducting a digital survey and focus group discussions. Some of the smoke-flavoring techniques, like atomized purified condensed smoke, is considered healthier than conventional smoking. Manufacturers of CSS are, however, hesitant to use all kinds of smoke-flavoring due to expected consumer skepticism. In the digital survey, the expected skepticism was confirmed (*n* = 487). Only 15% of the respondents were positively oriented toward smoke-flavoring of CSS. The selection criterion for the focus group discussion was based on the results from the survey and resulted in three attitude-oriented focus groups (negative, neutral, and positive). The negative focus group considered smoke-flavored CSS to be unnatural and out of the question to buy or eat. Unlike the negative focus group, the neutral group was curious to learn more and open to potential smoke-flavor benefits. However, credible information or science was crucial to influence CSS choice. Future studies may investigate whether the existent of this large group of curious neutral consumers (47% of the respondents) influence manufacturers (of CSS) skepticism or how various types of product information could steer consumer acceptance of smoke-flavoring of CSS.

## 1. Introduction

Cold smoked salmon (CSS) is a traditional Norwegian food product [1] often referred to as smoked salmon only. CSS is produced by salting, drying, and smoking fillets. These three processing steps preserve and flavor the product [2]. Conventional smoking, which is smoke obtained from smoldering wooden chips, is known to transfer chemicals such as polycyclic aromatic hydrocarbons (PAHs) and pollute the product and the environment with tar, ash, and soot [2,3]. A healthier and more environmentally friendly option for smoke processing of foods, is to use atomized purified condensed smoked (PCS) which is a smoke-flavoring technique [4]. PCS is generated on the basis of purified primary products and contains fewer harmful substances such as PAHs [4,5]. The European Parliament and the Council of the European Union supports the use of smoke-flavoring in food processing as it is considered to be healthier than conventional smoking [6]. However, Norwegian manufacturers of CSS are reluctant due to expected consumer skepticism influencing food choices and willingness to buy smoke-flavored CSS [7].

Waldenstrøm et al. [8] found no significant difference in consumers sensory acceptance between conventionally cold smoked salmon and a smoke-flavored variety. Similar perceptions were identified both by a trained sensory panel using Descriptive analysis (only 3 out of 23 attributes differed) and a consumer panel performing Check All That Apply (only 3 out of 29 terms differed). Although the study resulted in a promising smoke-flavored prototype with similar sensory- and physiochemical quality and consumer acceptance to conventionally cold smoked salmon, only uninformed acceptance and sensory perception were tested [8]. Consumers were not given any information about smoking protocol (smoke-flavored or conventionally smoked) and no immediate attitudes or believes were explored.

Consumers food choices are based on acceptance, liking, and sensory characteristics, but also related to cognitive factors [9,10], psychological factors, and theories concerning culture and practices [11,12]. Köster [10] states that combining various factors provides a better basis for consumer understanding, which highlights the need for interdisciplinary science on consumer response and behavior [13]. A frequently used method in social science is phenomenological analysis focusing on understanding the essence of an experience through description of a phenomenon while setting personal assumptions aside [14,15]. Nevertheless, the researcher’s own preconceptions will unconsciously become part of the dynamic interpretation process, which makes it important that several researchers, with various perspectives, interpret the data material, and present their views on the phenomenon in question [14]. The data material can, for example, be obtained via depth interview [16,17,18], focus group discussion [19,20,21,22], or observation [23,24].

According to Guerrero et al. [25], combining consumer insight and sensory knowledge will provide a comprehensive consumer understanding, which is especially useful in the new product development necessary to ensure food safety and food security for the future. Budhathoki et al. [22] found that providing information about production method significantly increased consumer liking of wild-caught CSS, when conducting a mixed-method study, combining surveys, sensory evaluations, and focus group discussions. 

The aim of the present study was to investigate whether the expected consumer skepticism was genuine by exploring Norwegian consumers’ attitudes and beliefs toward smoke-flavor processing of CSS. 

## 2. Materials and Methods

The study was organized into two steps. First, a digital quantitative survey to identify immediate attitude to smoke-flavoring. Second, three qualitative focus group discussions (phenomenological approach) to gain a deeper insight into the consumers attitudes and beliefs. To avoid confusion, CSS was defined and described as smoked salmon, using pictures and text, to the participants in both steps of the study. Conventional smoke processing was described using smoke from smoldering wooden chips, usually from deciduous trees or junipers. Smoke-flavoring was described as adding smoke concentrate in liquid form (dipping, brushing, spraying, condensing) to flavor and preserve the salmon.

### 2.1. Digital Survey

A digital survey was conducted in June 2020 by 487 Norwegian consumers, 18+ years old, who consumed CSS at least once a year. The survey was distributed in social media (Facebook) using the “snowball principle”. The questions were asked in Norwegian and were translated and back-translated [26], by the authors of this paper. The data were captured and anonymized using the software EyeQuestion^®^ (EyeQuestion^®^, Version 4.11.61, Provincie Gelderland, The Netherlands). 

The questionnaire included 29 claims regarding buying and usage patterns (9 claims), health perceptions (6 claims), and beliefs regarding CSS and smoke-flavoring (14 claims). The claims were discussed and selected based on The Food Choice Questionnaire [27], The Health and Taste Attitude Questionnaire [28], The Health Consciousness Scale [29], and the electronic questionnaire by Vanhonacker et al. [30]. The degree of agreement to each claim were measured on a 5-point Likert scale [31], ranging from “Disagree strongly” to “Agree strongly” with “Neither disagree nor agree” in the middle. All claims are listed in Appendix A, Table A1. 

### 2.2. Focus Groups

Potential participants were recruited digitally by the authors of this paper (Facebook). All candidates were asked to scale their agreement to the claims regarding smoke-flavoring of CSS included in the digital survey conducted 10 months earlier (Appendix A, Table A1). The attitudinal grouping explained in Section 2.3 was conducted soon after the completion of the digital survey and prior to the focus group interviews. The same global score limits as in the digital survey were used to establish a negative, a neutral, and a positive focus group. Due to the COVID-19 pandemic, the focus group discussions were conducted digitally in Microsoft^®^ Teams and the group size was set to five participants based on experiences from digital supervising and teaching as well as recommendations from literature [14,32]. The three qualitative focus group discussions were conducted in April 2021. All participants were 18 years or older eating CSS at least once a year. An overview showing the composition of the focus groups, with global score range for each group, is visualized in Table 1.

Due to a last-minute change, one of the participants in the negative group had a higher global score than planned (2.7).

#### 2.2.1. Protocol and Pre-Test 

A semi-structured approach was used, and a series of core topics were addressed. The participants were allowed to discuss other aspects that they also found relevant. The core topics were predefined in four themes: Attitudes and beliefs, Science, Naturalness and health, and Purchasing behavior. Questions and keywords were designed and selected based on guidelines from literature, the digital questionnaire, and previous studies [14,21,33,34] (Appendix B, Table A2). The protocol was pre-tested in a pilot focus group (*n* = 4) to check if the flow of questions was good, if the time frame was right, if any questions or keywords should be changed, and if the digital implementation was effective. The moderator and the observers got to practice their roles and interactions in the pre-test, which was perceived as an advantage. After the pre-test, a few keywords were added only to be introduced if the conversation stopped or drifted in a direction outside the scope. 

The focus group discussions were directed by the moderator and two observers and note takers. All sessions were video recorded and lasted for approximately 1 h. The participants were informed that there were no correct or incorrect answers, and that personal opinions and beliefs were of interest. The moderator was responsible for facilitating the group discussions and started by introducing herself before asking some simple and specific warm-up questions such as why the participants had agreed to join, whether they had participated in focus group discussions before, and their relationship to CSS. In the end of the focus group discussion, some minutes were set aside to give the participants the opportunity to summarize and add comments about the phenomenon (smoke-flavoring of salmon). The discussions were held in Norwegian and were translated to English after coding.

#### 2.2.2. Researcher’s Assessments of Validity and Reliability 

Our personal attitudes may have influenced the study, even though we strived for a neutral and objective approach. Some of the participants in the present study knew the moderator or one of the observers. According to Tjora [32], a close relationship can affect validity. To counteract such an influence, the moderator got everyone involved in the discussion from the very beginning and balanced each participants’ contributions. 

Another perspective is whether the results would have been the same if conducted by other researchers. The moderator and the two observers are food scientists and were familiar with smoked CSS in terms of smoking protocol and traditions. To ensure that the protocol was clear to the participants, it was tested in a pre-test and independent interpretations of the findings were used (the essence was assessed by several authors). For validity reasons, we repeatedly went back to the transcripts and evaluated our coding and condensation to avoid losing the essence. 

According to Creswell and Poth [14], reliability is, for example, about choosing the right method and asking the right questions. In this study, our interests were to explore the attitudes and beliefs of three attitudinal consumer groups. It seemed obvious to choose a qualitative method focusing on the phenomenon in question, to gain additional insight into the participants’ understanding and attitudes towards smoke-flavoring of CSS.

### 2.3. Attitudinal Grouping and Data Processing 

Seven of the claims from the digital survey, Table 2, were considered suitable to measure the immediate attitude or response to smoke-flavoring of CSS. Negatively phrased claims were reversed before summing and individual global scores were derived from the seven claims. Furthermore, attitudinal groups (negative, neutral, and positive) were formed based on individual global scores. The negative group consisted of participants with global scores ≤ 2.5 and the positive group consisted of those with global scores ≥ 3.1. The rest of the participants belonged to the neutral group (global scores 2.6–3.0).

Factor Analysis (Principal Component Analysis with Varimax rotation) for the attitudinal grouping was performed using IBM^®^SPSS^®^ Statistics software (IBM^®^ SPSS^®^, release 28, Endicott, NY, USA). Two components were extracted and the factor loadings for all claims scored in the same direction for the first component explaining 30.91% of the total variance. The Kaiser’s criterion, eigenvalue ≥ 1 for component extraction, and Cronbach’s alpha value ≥ 0.7, for internal consistency and reliability, was fulfilled [35,36].

To characterize the attitudinal groups, partial least squares regression (PLS-R) with a backward stepwise method was used. First, sociodemographic characteristics and all claims, except the seven latter questions regarding smoke-flavoring of CSS (Appendix A, Table A1), were included as independent variables (X) to check which ones were related to each attitudinal group (Y). Then, several refined models were run by selecting only the significant variables, as independent variables (X). In all models, standardized X-variables, cross-validation with 20 random segments, and significance testing by jack-knifing at 5% significance level were used [37]. The models were run in Aspen UnscramblerTM V. 12.1 (Aspen Technology Inc., Bedford, MA, USA).

The data from the focus group discussions were anonymized and analyzed using a stepwise phenomenological approach: familiarization, identifying and sorting meaning units (coding), detecting the essence of the phenomenon (text condensation), and developing textural and structural descriptions (synthesizing) [14,15]. The transcripts were coded by the two observers and the moderator separately using NVivo 1.5.1 (QSR International, Burlington, VT, USA).

## 3. Results

### 3.1. Digital Survey

The attitudinal grouping of the 487 participants, based on the individual global score limits (Section 2.3), is presented in Table 3. The negative group consisted of 37.8% of the participants while 15.0% were included in the positive group. The rest of the participants i.e., 47.2%, belonged in the neutral group.

The sociodemographic characteristics in total and per attitudinal group are presented in Table 4. The way the participants were recruited resulted in a compositional bias compared to the Norwegian population [38,39,40,41].

The purpose of the digital survey was to measure immediate attitude toward smoke-flavoring and give an overall description of the consumers belonging to the three attitudinal groups. A large proportion of the respondents were either negative or neutral towards smoke-flavoring of CSS, but the neutral consumers were poorly described by the PLS-R model. The final one-factor PLS-R models (data not shown) explained 13%, 2%, and 8% of the variation in immediate negative, neutral, and positive attitude toward smoke-flavoring of salmon, respectively. This indicated that the selected variables only partly characterize the positive consumers and were unable to characterize the neutral consumers. To visualize the descriptions of the consumers belonging to the negative and positive attitudinal groups, a plot including both attitudinal groups as dependent variables, was prepared (Figure 1). 

The participants with a negative attitude towards smoke-flavoring of CSS were typically older than 40 years and less educated than the participants with a positive attitude (Figure 1). They were concerned with food naturalness, agreeing to the claims regarding the importance of natural ingredients, no additives, and no artificial ingredients in everyday food. The claims “Choosing the “right piece” is important to me when I purchase smoked salmon” and “Not farmed (wild) smoked salmon is better than farmed smoked salmon” were agreed upon too. The participants with a negative attitude towards smoke-flavoring of CSS did however disagree to the claims “Low price is important to me when I buy smoked salmon”. 

The participants with a positive attitude towards smoke-flavoring of CSS were typically more educated and younger than the participants with a negative attitude. They agreed to the claim “Low price is important to me when I buy smoked salmon” and disagreed to the claims regarding the importance of food naturalness and that CSS made of wild (not farmed) fish is better than the ones made from farmed salmon.

### 3.2. Focus Groups

All focus groups had lively discussions, even though they participated digitally. The themes from the protocol (Appendix B, Table A2), categories, and codes are organized and visualized in Table 5. The findings from each category are summarized separately and in the same order in all subsections. First, the findings from the negative group, then the findings from the neutral group, and, finally, the findings from the positive group.

#### 3.2.1. Familiarity and Curiosity

The negative group believed smoke-flavoring to affect healthiness and sensory perception negatively. They were not interested to learn more about smoke-flavoring, as it was not relevant to buy, eat, or use. The sensory perception and tastiness were highlighted as the most important factor in choosing CSS and in their opinion smoke-flavored CSS, was a second-class product.


*… It is second class and doesn’t taste as good. If I had been told that smoke-flavor was used, I would never have bought it. Unnatural*


During the negative group’s discussion, industry benefits like efficiency, profit, and consistent quality, were highlighted. The group did not believe smoke-flavoring to have any consumer benefits and found smoke-flavor processing to violate the national “food smoking heritage”. A conscious and emotional relationship to conventional smoking was found in the negative group associating conventional smoking with artisanal food, which they respected and were proud of. They believed craft food production to be a time-consuming and knowledge-based art form.


*… And when you enter the store [local smokehouse], the odor is special, probably not pleasant to everyone. The experience getting in there…, you meet like hundreds of years of fine-tuning smoke processing and fish handling. And the experience of getting into such a store is incredibly nice.*


Both the neutral and the positive group believed smoke-flavoring to be more efficient, profitable, and environmentally friendly than conventional smoking and favorably to ensure consistent quality. Conventional smoking felt safe to the neutral group, even though they had not thought much about it in the past.


*I haven’t really thought about various ways of smoking salmon, but on the package, I sometimes read “juniper smoked” and sometimes other things…, ehm.*


Although the neutral group believed smoke-flavoring to be beneficial for several reasons, the group questioned the need to change a perfectly functional way of producing CSS, namely conventional smoking. Still, they were curious about smoke-flavoring believing there were some benefits they did not know of, possibly related to environment or health.


*I am curious about smoke-flavoring, but I do not know what it is. What benefits do smoke-flavoring have—compared to conventional smoking?—environment, health?*


The positive group perceived smoke-flavoring to be “fake”, like sugar-free soda, and “deceiving” used by the industry. The group believed equally high quality, compared to conventionally smoked CSS, was important in smoke-flavor processing to avoid disappointed consumers. They believed conventional smoking to be the traditional way of producing CSS using descriptors like small-scale, traditional, and craft production. 


*I associate conventional smoking with old processing skills. It’s not a modern thing. Maybe conventional smoking is a bit out of the picture in modern production? But… yes, I get associations to the old days.*


Simultaneously, the group believed that a high share of the consumers would prefer to buy the known (conventionally smoked CSS) over the unknown (smoke-flavored CSS).


*… So, the critical consumers… If they find out that this is not conventionally smoked, they will deviate from it, and rather buy another product.*


#### 3.2.2. Trust in Science

All groups trusted science in general, but were skeptical of industry beneficial science assuming incomplete and selective research. Ulterior motives and intentions were considered important to be able to assess credibility. 

The negative group considered subjective established beliefs to be the most important trust influencer. Science agreeing with one’s personal opinions is easier to accept than science contradicting individual beliefs. To change or influence established beliefs, well-considered arguments and documentation are required.


*But, if I’m super engaged and involved in something and some research claims the opposite of what I believe, it’s not easy to convince me to change my opinion... Then I really need to know where the research is coming from and if it’s trustable or not.*


The neutral group believed confirmatory documentation from independent research groups strengthened credibility and trustworthiness, regardless of the research area, e.g., science submitted by government authorities, like the Norwegian Food Safety Authority (Mattilsynet).


*Yes, I trust almost all science, except research coming from the industry itself. And yes, I am aware that some research projects at universities and university colleges are sponsored by the industry. Whether I trust “industry-sponsored science” or not depends on the amount of industry influence (Is science bought by industry?), I am able to detect.*


Although the neutral group highlighted the importance of being source-critical, they admitted not always being so. 


*… But it may be like... I read something, like an article, and even if it’s just one person’s view or text, I end up accepting the content—Is that so? Okay*


The positive group was not particularly influenced by research area but emphasized the importance of being source-critical and aware of the fact that no research is “complete”. Simultaneously, they admitted being affected by blog posts and influencers even though they described this information as untrustworthy.

#### 3.2.3. Health

The negative group considered smoke-flavor to be an unhealthy, unnatural, and carcinogenic additive. The group was able to believe that smoke-flavoring could be healthier than conventional smoking, simultaneously stating that proven health benefits would not be enough to change their food choices. They relied on conventional smoking due to traditions and heritage, and believed the small amounts of CSS they consumed were not harmful anyway. 


*If you do not eat smoked salmon every day, 365 days a year, in large quantities, I do not think it is harmful to choose conventionally smoked, no matter what they say.*


The neutral group associated smoke-flavoring with something unknown and alarming while the positive group was aware of the harmful substances in smoked foods and believed smoke-flavoring to be an interesting alternative if harmful substances could be removed or reduced. To believe this, credible research had to be presented, proving that smoke-flavoring was healthier than conventional smoking. Despite this, conventionally smoked CSS “sounded healthier” than smoke-flavored CSS, in their opinion.

#### 3.2.4. Food Naturalness

The negative group was quite clear, conventional smoking was considered natural; smoke-flavoring was not. The use of additives was emphasized by the negative group, trying to explain food naturalness. They understood that additives could be useful, but were critical of their use, and considered additives to be unnatural and to influence food choice negatively. The neutral group defined food naturalness as unprocessed, clean, and healthy foods directly from farms or the sea. They considered Norwegian food to be more natural than imported food and believed that a Norwegian appearance, label, or brand would strengthen the idea, or feeling, of naturalness.


*I think I am easily fooled if the product expresses “from Norway” or “made in Norway” e.g., by label or the way it’s wrapped. It feels safe and natural to me.*


The neutral group believed conventional smoking to be natural, but expressed lacking the knowledge to be able to assess the naturalness of smoke-flavoring. The positive group discussed food naturalness vividly and considered it to be related to health, degree of processing, and the use of additives. If all processing changed the naturalness e.g., if only natural ingredients were included, was unclear to the positive group. It was highlighted that the term “natural” is being misused by the food industry, as an advertising “label”, to justify high prices.


*Yes, I think we consider natural foods as foods made from few ingredients. Meat is natural, but if you mix meat, onions, and lard in an intestine—make a sausage—then it is no longer natural. Hmm… but every ingredient is natural...*


The positive group immediately replied that conventional smoking is natural, and smoke-flavoring is unnatural. In further discussions, it was indicated that smoke-flavor processing probably consisted of natural ingredients, and could be natural, nonetheless.

#### 3.2.5. Choice Awareness

If sensory quality and price were equal, all groups would have bought conventional CSS. During the discussions, it emerged that in order to influence CSS choice or preference, proven smoke-flavor benefits or substantial price differences were required.

The negative group appeared to be alert consumers. When choosing CSS, they considered price, manufacturer, appearance, and packaging and preferred small-scale products, if available. They were skeptical of new techniques like smoke-flavoring, but believed it to be a good alternative for those mainly concerned with low prices. The negative group believed high-quality CSS and high price to be connected and considered smoke-flavored CSS to be cheap and of poorer quality.


*… But if I was going to buy CSS, on a rare occasion like contributing to a May 17 (National day) buffet, I would not have bought the cheapest one… I had assumed it wasn’t good enough.*


The positive group highlighted the importance of smoke-flavor advantages, such as health, taste, or environment, to influence consumer choice. Good sales arguments were crucial to change consumers’ attitudes. The group was quite aware of their food choices and believed personal gain was necessary to be able to change their choices and habits.


*If it’s because it’s going to be faster or easier to produce, then it’s not that important to me. I don’t really care if the manufacturers struggle to make my food. It’s not that important to me. But if a lot of potatoes disappear in the process, or fish suffer unnecessarily, or the climate impact is greater, then it is important to me.*


The positive group assumed that smoke-flavoring was applied to produce, for example, sausages, smoked mackerel, and salami. Smoke-flavoring of these products was unproblematic and accepted by the group, unlike the production of traditional foods, like CSS. The neutral group believed that their choice awareness would be strengthened by increased marketing and information flow. They found Norwegian brands and well-known manufacturers to be trustworthy. It was commented, by the group itself, that the confidence in Norwegian manufacturers was more comprehensive than previously commented in Section 3.2.2. 


*So, if a well-known producer informs me about a new “smoking” method, I would probably have accepted it—Because I trust them.*


Acting according to habits, not always being aware of personal food choices, was mentioned by the neutral group believing the trust in Norwegian manufacturers could be due to unawareness and routine. We trust them until evaluating trustworthiness.


*I had probably grabbed the same CSS as always. Just unconsciously picked the CSS I usually buy.*


## 4. Discussion

This study aimed to explore Norwegian consumers’ skepticism and attitude towards smoke-flavoring of CSS. The present study found a genuine and immediate skepticism among the participants. A substantial part (38%) of the participants were negative, while only 15% were positive towards smoke-flavoring of CSS. The largest group of participants (47%) were defined having a neutral attitude toward smoke-flavoring of CSS. However, the participants in the neutral group had global scores between 2.6 and 3.0 which might be interpreted as a slightly negative attitude. 

Consumers in general are skeptical towards new or unknow technology (i.e., irradiation or biotechnology), even if the overall acceptance seems to vary with type of innovation. Albertsen et al. [42] stated that some recent technology-based innovations have high consumer acceptance (e.g., enriched or improved functional food), while others are more often rejected by consumers (e.g., genetically modified food). Although smoke-flavored CSS is neither irradiated nor produced using biotechnology, the skepticism detected in the focus group discussions was similar to the skepticism detected by Albertsen et al. [42]. Guerrero et al. [25] highlighted that Norwegian consumer are critical to food innovations, especially when it comes to traditional food products. This is in line with the skepticism towards smoke-flavor processing, discovered in the present study. Conventional smoking was perceived to keep old artisan traditions alive, and the need to change the traditional way of processing were questioned by the neutral focus group. The fact that young and educated people tend to be more open to new food processing technology is known from previous studies [43,44,45,46] and confirmed in the present study. 

Results from the survey and the focus group discussions clearly highlighted some attitudes and barriers towards CSS, especially among the negatively oriented participants. The main concerns were the perceived unnaturalness, the lack of healthiness, and the expected reduction in sensory quality. In addition, a negative attitude towards CSS made from farmed fish was detected. Other issues revealed were the importance of potential benefits, curiosity, and trustworthiness.

### 4.1. The Issue of Unnaturalness and Lack of Healthiness

The most pronounced correlation for the consumers with an immediate negative attitude towards smoke-flavoring of CSS, was the high agreement to the three claims regarding food naturalness indicating high food naturalness importance [47]. This is consistent with Waldenstrøm et al. [8] detecting the presence of the attribute “natural” to be an important driver for consumer liking of CSS. The participants in the negatively oriented focus group considered conventional smoking to be natural, and smoke-flavoring to be unnatural. The participants in the two other focus groups expressed lacking the knowledge to really assess the naturalness of smoke-flavoring, even if they believed that most people would consider conventionally smoked CSS to be more natural than smoke-flavored CSS. Román et al. [47] classified food naturalness into categories dealing with technology, food origin, and ingredients as well as considering natural as a product attribute. A similar classification was conducted by the participants in the neutral and positive fucus groups in the presents study. Mildly processed or unprocessed food from Norway, directly from farm or sea, made from “clean” ingredients with no additives, was considered natural. Another factor, brought up during the focus group discussions, was the relationship between healthiness and food naturalness known from previous studies [47,48,49]. 

The participants in the negatively oriented focus group considered smoke-flavor to be an unhealthy, unnatural, and carcinogenic additive and the participants in the neutral focus group associated smoke-flavoring with something unknown, alarming, and unnatural. The participants in both focus groups were able to believe that smoke-flavoring could be healthier than conventional smoking. If this fact were proven, it still would not change the food preferences for the participants in the negative group. They relied on conventional smoking due to traditions and heritage and believed the small amounts of CSS they consumed, were not harmful anyway. The fact that even information concerning health is unable to change food choice is interesting. This may indicate unawareness or ignorance about the health risks associated with consuming mildly processed seafood [50], the carcinogenic effect of smoke processing [2,51], or an already established belief that conventional smoking is somehow better.

### 4.2. Sensory Perception and Wild Fish vs. Farmed

Sensory perception and taste were highlighted as the most important factors when choosing CSS, by the negatively oriented focus group. In their opinion, smoke-flavored CSS do not taste as good as conventionally smoked CSS. Waldenstrøm et al. [8] found no significant difference in consumer acceptance between conventional CSS and a smoke-flavored variety which may indicate that tasting the products, while explaining the potential smoke-flavor benefits, could positively influence attitude. To succeed in this, credible science or documentation must be presented. It may be wise to focus on the big group of consumers with an immediate neutral attitude (47%) instead of the negatively oriented consumers with fixed established beliefs that will take a lot of effort to influence in any direction. 

The participants with the immediate negative attitude towards smoke-flavoring of CSS were to a greater extent in agreement with the claim that CSS made from wild salmon is better than CSS made from farmed salmon, than the participants with the immediate positive attitude towards smoke-flavoring of CSS. Previous studies have found that the perceived naturalness and healthiness are two of the reasons why people choose wild-caught salmon over farmed salmon [22,52]. Budhathoki et al. [22] found that the overall liking (92 Danish consumers) of wild-caught smoked salmon significantly increased after production method information was provided (wild-caught) and stated that factors such as beliefs, norms, and attitudes influence consumer preference for wild and farmed fish. 

The participants in the negative focus group believed high-quality and high price were connected and considered smoke-flavoring to produce cheaper CSS of poorer sensory quality. To them, high-quality concurred low prices, especially when buying CSS occasionally. Like previously mentioned, CSS is considered to be a traditional food product in Norway and previous studies have shown a higher price acceptance for traditional foods [1,53]. However, the positively oriented participants in the present study believed low price to be important when buying CSS and highlighted the need for a substantial price difference or other proven smoke-flavor benefits, like environment or health, in order to positively influence the buying frequency of smoke-flavored CSS.

### 4.3. Benefits, Curiosity, and Trustworthiness 

The participants in the negative focus group did not believe smoke-flavoring to have any consumer benefits, but several industry benefits like efficiency, profit, and consistent quality. The industry benefits did not influence consumer attitude positively, in their opinion. Vanhonacker, Kühne et al. [54] also experienced that consumer acceptance was strongly influenced by the perceived impact of the innovation, dealing with traditional foods, like CSS. 

The participants in the neutral focus group were curious about smoke-flavoring, believing there were some benefits they did not know of, possibly related to environment or health. This curiosity clearly separated the groups. The participants in the neutral focus group were open to “unknown” (to them) smoke-flavor benefits and were curious to learn more. The participants in the negative focus group were not interested in learning more about smoke-flavored CSS, as it was not relevant for them to buy, eat, or use. Curiosity is known to be a motivational drive for seeking knowledge [55,56], catalyzing the willingness to try novel foods [57], and increasing the acceptance of new food processing technology [58]. 

During the focus group discussions, it emerged that the intention behind the information was important to be able to assess trustworthiness. The participants, in all three focus groups, trusted independent science, in particular science submitted by government authorities like the Norwegian Food Safety Authority (Mattilsynet). However, they were all skeptical of industry beneficial science assuming incomplete and selective research. In Norway, basic trust in authorities is high [59], while consumer attitude towards the food industry is influenced by multiple factors. According to Thorsøe et al. [60], there is a strong link between credibility and trust. The manufacturers must be credible to be trusted by the consumers. Trying to control consumers’ perceptions or attitude will only create mistrust [60]. 

It was commented, by the participants in the neutral focus group, that the confidence in Norwegian manufacturers was high, partly because they trusted the Norwegian food industry and partly due to ignorance or unawareness. It could be that the confidence in the Norwegian food industry indirectly is affected by the trust in the Norwegian Food Safety Authority trusting them to control the industry.

The participants in the negative focus group considered subjective established beliefs to be an important trust influencer and pinpointed that well-considered arguments and documentation were required to be able to change an established belief. This is line with the findings of Llauger et al. [61], detecting reliable information on sensory and health-related issues to be crucial to alter people’s subjective norms (i.e., people’s personal beliefs influenced by the perceived expectations from others). The same need for documentation was not found among the participants in the positive focus group. They admitted being affected by blog post and even influencers, even though they knew that more comprehensive information was available. The reason for trusting these sources could be to minimize the cognitive effort [62,63], and not always have to take a well-considered stand. 

### 4.4. Limitations and Future Work

Overall, the attitudinal grouping based on the seven claims were not well explained by the PLS-R models. There might be several reasons for this (i.e., questionnaire design, imbalanced consumer groups, etc.); however, a possible explanation is the seven claims itself. No literature on how to measure the immediate attitude towards CSS is known to the authors, and measuring attitudes is generally known to be difficult [64,65]. The reason for choosing these exact seven claims was to include key aspects such as health, sensory perception, perceived naturalness, additives, safety, environment, and willingness to buy CSS. Although a low explained variance was seen, some tendencies and characteristics were revealed. 

The focus group discussions were conducted digitally in Microsoft^®^ Teams due to the COVID-19 pandemic. The group size was set to five participants based on experiences from digital guidance and teaching as well as recommendations from literature [14,32]. Nevertheless, the five-group size was in the lower part of the recommendations. To vali-date the findings, several interviews could have been conducted in each attitudinal group (with new sets of consumers) or the group size could have been increased slightly. 

In future research, it would be of great interest to investigate whether the existence of this large group of curious consumers (the neutral group), who seem open to credible information about smoke-flavor benefits and the fact that there were no significant differences in consumer acceptance between conventional and smoke-flavored CSS [8], affects the Norwegian manufacturers’ (of CSS) skepticism. 

To investigate how knowledge about smoke-flavor processing could affect attitude, especially in the curious neutral group, would be of great interest too.

## 5. Conclusions

The aim of this study was to investigate whether the expected consumer skepticism was genuine by exploring Norwegian consumers’ attitudes and beliefs toward smoke-flavor processing of CSS. 

The digital survey revealed a prominent consumer skepticism. Only 15% of the respondents were positive toward smoke-flavoring of CSS. The participants with a negative attitude were typically older and less educated than the participants with a positive attitude toward smoke-flavoring of CSS. The negatively oriented participants were concerned with food naturalness and believed CSS made from wild salmon was better than CSS made from farmed salmon. 

The findings from the focus group discussions clearly highlighted some attitudes and barriers towards smoke-flavoring off CSS. The main concerns were the perceived unnaturalness, the lack of healthiness, and the expected reduction in sensory quality. Other issues revealed were the importance of potential benefits, curiosity, and trustworthiness. Industry benefits were not perceived as drivers for accepting smoke-flavoring, while health- or environmental benefits were referred to as possible drivers for acceptance, especially for the big group of curious consumers (the neutral group). It is worth highlighting that the large group of neutral consumers (47% of the participants in the digital survey) were slightly skeptical towards smoke-flavoring of CSS, but curious to learn more and open to documented smoke-flavor benefits. The participants in all three focus groups trusted independent science, particularly science submitted by government authorities like the Norwegian Food Safety Authority (Mattilsynet), but were skeptical of industry beneficial science assuming incomplete and selective research. 

This study shows that it may be wise to combine quantitative and qualitative methodology to bring out the various aspects of consumers’ attitudes and beliefs. The first part of the study, the digital survey, used quantitative methodology to explore the “Whats” (What is the immediate attitude and what characterize the attitudinal consumer groups?), while the focus group discussions used qualitative methodology to explore the “Whys” (Why do the attitudinal focus groups have their attitudes and beliefs?).

## Figures and Tables

**Figure 1 foods-11-02170-f001:**
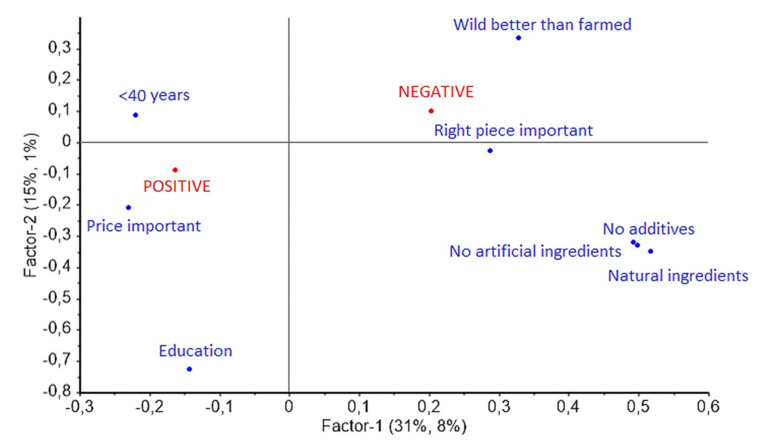
Sociodemographic characteristics, general beliefs and beliefs regarding cold-smoked salmon characterizing the negative and the positive attitudinal group in a PLS-R model.

**Table 1 foods-11-02170-t001:** Characteristics of the focus groups with global score ranges ^1.^

Group	Number of Participants	Gender		Age Range	Global Score Range ^1^
		Male	Female		
Positive	5	2	3	21–60	4.1–3.3
Neutral	5	1	4	21–70	3.0–2.6
Negative	5	1	4	21–60	2.7–2.0

^1^ Based on global scores of seven claims regarding smoke-flavoring of CSS evaluated on 5-point Likert scales.

**Table 2 foods-11-02170-t002:** Claims for measuring immediate attitudes towards smoke-flavoring of CSS.

Claims
I believe smoke-flavoring of salmon is as healthy or healthier than conventional smoking
I believe smoke-flavoring does not affect the taste experience
Smoke-flavoring of salmon is not natural ^1^
Smoke-flavoring of salmon is not safe ^1^
Smoke-flavoring of salmon is better for the environment
Conventionally smoked salmon contains fewer additives than smoke-flavored salmon ^1^
If smoke-flavored salmon were available in the grocery store, I would have bought it

^1^ Indicates reversed items for the computation of smoke-flavor attitude in CSS production.

**Table 3 foods-11-02170-t003:** Immediate attitude ^1^ grouping towards smoke-flavoring of CSS with mean global scores and standard deviation (SD) (*n* = 487).

Attitudinal Groups	Negative (*n* = 184)	Neutral (*n* = 230)	Positive (*n* = 73)
	Mean (SD)	Mean (SD)	Mean (SD)
Immediate attitude towards smoke-flavoring of CSS ^1^	2.03 (0.38)	2.82 (0.17)	3.33 (0.23)

^1^ Based on global scores of seven claims regarding smoke-flavoring of CSS evaluated on 5-point Likert scales.

**Table 4 foods-11-02170-t004:** Sociodemographic characteristics, in total and per attitudinal group, of the participants in the digital questionnaire (*n* = 487).

Sociodemographic Characteristics	Total *n* (%)	Negative *n* (%)	Neutral *n* (%)	Positive *n* (%)
Gender ^1^				
Male	130 (26.7)	51 (27.7)	52 (22.6)	27 (37.0)
Female	357 (73.3)	133 (72.3)	178 (77.4)	46 (63.0)
Age Group				
<40 years	138 (28.3)	39 (21.2)	75 (32.6)	24 (32.9)
41–50 years	197 (40.5)	80 (43.5)	86 (37.4)	31 (42.5)
>50 years	152 (31.2)	65 (35.3)	69 (30.0)	18 (24.7)
Education ^1^				
Lower secondary school	6 (1.2)	3 (1.6)	2 (0.9)	1 (1.4)
Higher secondary school	49 (10.1)	27 (14.7)	18 (7.8)	4 (5.5)
Technical college degree	41 (8.4)	15 (8.2)	23 (10.0)	3 (4.1)
Bachelor’s degree	180 (37.0)	73 (39.7)	83 (36.1)	24 (32.9)
Master’s degree	148 (30.4)	51 (27.7)	75 (32.6)	22 (30.1)
Doctorate	53 (10.9)	6 (3.3)	28 (12.2)	19 (26.0)
Total income in household before deductions and taxes (NOK/year)				
<599,000	65 (13.3)	20 (10.9)	34 (14.8)	11 (15.1)
600,000–999,000	140 (28.7)	55 (29.9)	67 (29.1)	18 (24.7)
1,000,000–1,399,000	146 (30.0)	51 (27.7)	73 (31.7)	22 (30.1)
>1,400,000	117 (24.0)	52 (28.3)	49 (21.3)	16 (21.9)
Do not know or wish to answer	19 (3.9)	6 (3.3)	7 (3.0)	6 (8.2)
Persons in household (including you)				
One	51 (10.5)	17 (9.2)	28 (12.2)	6 (8.2)
Two	135 (27.7)	51 (27.7)	62 (27.0)	22 (30.1)
Three	86 (17.7)	32 (17.4)	41 (17.8)	13 (17.8)
Four	142 (29.2)	59 (32.1)	65 (28.3)	18 (24.7)
Five or more	73 (15.0)	25 (13.6)	34 (14.8)	14 (19.2)
Under aged children (<18) in household				
None	202 (41.5)	79 (42.9)	96 (41.7)	27 (37.0)
One	88 (18.1)	33 (17.9)	43 (18.7)	12 (16.4)
Two	137 (28.1)	53 (28.8)	61 (26.5)	23 (31.5)
Three or more	60 (12.3)	19 (10.3)	30 (13.0)	11 (15.1)
Main purchaser of food and drink in household ^1^				
My parents	2 (0.4)	2 (1.1)	0 (0)	0 (0)
My partner	35 (7.2)	9 (4.9)	11 (4.8)	5 (6.8)
We share equally	208 (42.7)	87 (47.3)	109 (47.4)	31 (42.5)
Me	242 (49.7)	86 (46.7)	110 (47.8)	37 (50.7)
Main occupation ^1^				
Disabled, not able to work	17 (3.5)	10 (5.4)	6 (2.6)	1 (1.4)
Unemployed, on leave or retired	23 (4.7)	10 (5.4)	9 (3.9)	4 (5.5)
Studying (Student, pupil, apprentice)	17 (3.5)	6 (3.3)	8 (3.5)	3 (4.1)
Employed (full or part time)	430 (88.3)	158 (85.9)	207 (90.0)	65 (89.0)

^1^ Included the option “other”, with the possibility to comment. For gender, “other” were not ticked, for the other characteristics, the comments were used to place the answers into the right categories.

**Table 5 foods-11-02170-t005:** Themes, categories, and codes.

Themes	Categories	Codes
Attitudes and beliefs	Familiarity and curiosity	Fake
		Efficient
		Cost effective
		Environment friendly
		Violates traditions
		Artisan food
		Special occasions
		Curiosity
		Skepticism
	Health	Healthiness
		Harmful
		Carcinogenic
	Sensory quality	Sensory perception
		Quality vs. price
		Taste
		Appearance
Science	Trust in Science	Intentions
		Credibility
		Research area
		Source-criticism
		Effort
		Established beliefs
		Engagement
		Industry
		Food Safety Authority
Naturalness and health	Food Naturalness	Processing
		Ingredients
		Health
		Origin (farm or sea)
		Local
		Additives
		Artificial
		Unnatural
		Packaging
Purchasing behavior	Choice awareness	Advantages
		Habits
		Unknown
		Norwegian
		Brand
		Manufacturer
		Small-scale
		Preference
		New technology
		Appearance
		Price importance

## Data Availability

Raw data are available on request.

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
