# Peer review of "Norwegian Consumers’ Skepticism towards Smoke-Flavoring of Salmon—Is It for Real?"

_foods, 2022, doi:10.3390/foods11142170_

Round 1

Reviewer 1 Report

Lines 71 to 76: this information should be moved to material and methods section. Also, it must be rewritten not to present results. E.g.: First, a digital quantitative survey was done to identify immediate attitude to smoke-flavoring. Second, qualitative focus group discussions (phenomenological approach), were carried out to gain a deeper insight into the consumers attitudes and beliefs identified.

Lines 77 to 83: this paragraph belongs to the introduction section. It could be move to the first paragraph, before the cold smoke salmon explanation.

Line 85: Please, add the sociodemographic info of the survey here (lines 113-114 and table 3).

Line 87: typo error “snowball principle”.

Line 97: please, indicate the total number of items of the questionnaire, and the umber of items of each part.

Lines 103 to 108, and table 2: Those results should be placed in the results section.

Lines 119 to 125: this information is redundant. Readers could find it in the table.

2.1.2. Statistical approach section: introduce a brief explanation about how to group consumers (terciles?). Use the “how” explanation included in lines 103-108.

Lines 144 to 146: move this information to the recruitment section (2.2.1), or move the sociodemographic information to this paragraph.

2.2.2 Protocol and pre-test: participants received an informed consent?

Line 215: I miss an explanation about the data variance explained in the PLS-R. Maybe the neutral consumers were not included due to the analysis excluded a large amount of data.

Lines 245 to 247 are not necessary.

3.2. Focus groups: I miss an scheme representing the main issues of each topic per consumers group to sum up all the information. I suggest authors to include it.

Author Response

The manuscript foods-1805844 (title: Norwegian consumers' skepticism towards smoke-flavoring of salmon – Is it for real?) is now revised according to the reviewers` comments. Below follows a list of changes or rebuttals according to each point raised by the reviewers. Our responses/comments are highlighted with a red font and all changes in the revised manuscript are marked using the “Track Changes” function. 

Reviewer 2 Report

Interesting work. However, the manuscript requires improvement as to defining the sections. Some information is required in the materials and methods is missing (see specific comments below). On the other hand, some information provided in the materials and methods section fits much better in the results section. In addition, the statistical approach (2.1.1)/data analysis (2.2.3.) sections for the survey and the focus groups should be presented in one section, not separately as it is affecting readability.

Please check some typos thorough the manuscript e.g. it should be revealed NOT reviled

Materials and methods

Line 99: What as the criteria to decide that the seven selected claims were suitable to measure the immediate attitude or response to smoke-flavoring of CSS? Can you please elaborate.

Line 92-93: More detail on the discussion and criteria for the selection of the claims is needed in the manuscript in the material and methods section.

Line 113-119: This section fits better in the results section.  Same as tables 2 and 3.

Line 130-131: more suited to results section.

Line 149: Please provide more detail of the focus group participants recruitment. What was the criteria/process to select these participants? Why only 15 participants for the focus groups?

Line 153: If the global score was one of the selection criteria for the focus groups it needs to be stated accordingly, otherwise this sentence and table 4 should be on the results section.

Line 192-208: Most of this section needs to go in the discussion section. If reading as limitations/shortcomings of this study and definitely not as methods.

Results

This section should start with an overall description of the surveyed participants the other results of the survey. Then followed by the focus groups results.

Author Response

(The authors gave the same response as above.)
